# Functionality of Native Starches in Food Systems: Cluster Analysis Grouping of Rheological Properties in Different Product Matrices

**DOI:** 10.3390/foods9081073

**Published:** 2020-08-06

**Authors:** Joanna Le Thanh-Blicharz, Jacek Lewandowicz

**Affiliations:** 1Department of Food Concentrates and Starch Products, Prof. Wacław Dąbrowski Institute of Agriculture and Food Biotechnology, Starołęcka 40, 61-361 Poznań, Poland; 2Faculty of Engineering Management, Poznań University of Technology, Jacka Rychlewskiego 2, 60-965 Poznań, Poland; jacek.lewandowicz@put.poznan.pl

**Keywords:** waxy starch, binary mixtures, salt, polyols, product matrix

## Abstract

Industrial application of starch as a texture-forming agent is primarily limited to preparations obtained from waxy corn and potatoes. The main reason behind this is its functionality, which depends mostly on rheological properties. However, in food product matrices, these properties change. Despite the vast amount of information on the rheological properties of various starches, the rational choice of thickener appears to be an extremely difficult task. The aim of the work is to systemize the information on the rheological properties of most popular starches in matrices of various food products, applying principal component and cluster analyses. The investigated material is potato and corn starch of the normal and waxy varieties. Binary mixtures containing salts or sweetening agents, as well as four different food products (ketchup, mayonnaise, pudding, and jelly), are investigated. It was found that compared to normal varieties, waxy starches reveal many similar rheological properties in all investigated models and food systems. Furthermore, in most applications, one waxy starch variety may be substituted by another, with no significant impact on the rheological properties and texture of the food product. Moreover, waxy starch preparations are less altered by the presence of cosolutes, i.e., salts and sugar alcohols. Starch model systems were proven to be useful only for rapid thickener screening tests and cannot be recommended as a final reference for the quality design of food products.

## 1. Introduction

The importance of starch in food production is not limited to be the main source of energy in human nutrition. It is the most popular texture-forming agent, which is related to its appropriate physicochemical properties, the safety of use, and low price. Starches of different botanical origins reveal a huge diversity in their structure and composition, including their amylose-to-amylopectin ratio and minor (lipids, proteins, and minerals) constituent content. As a consequence, they display various functional properties such as thickening capability, gelling or stabilizing ability, and transparency [1]. Moreover, due to the ability of starch to be physically and chemically modified, starch properties can be precisely tuned for specific applications in food systems [2,3,4]. It is also possible to change the nutritional properties of modified starch [5]. Methods and ranges for starch modification are precisely specified by Codex Alimentarius [6] to ensure the nutritional safety of these products. Nevertheless, the clean label trend observed in recent years increases the interest of food producers in the use of native, especially waxy, starches [7].

A number of different factors determine the functionality of starch as a food thickener and stabilizer. Nevertheless, analysis of its rheological properties is a primary method used for the examination of this biopolymer in both industry and scientific studies [8]. A number of different devices, with varying degrees of complexity and precision of measurements, are used for these analyses. Viscographs that reflect the technological conditions in which starch preparations are used, as well as rheometers with a defined shear rate, are particularly useful. The most technically advanced devices provide oscillatory tests that enable us to study the rheological properties of gels without the destruction of their structure. Unfortunately, they are not very popular and commonly used, while viscometers give results that are usually impossible to compare.

The most important technological information provided by rheological measurements are gelatinization temperature, viscosity, and rheological stability (in particular, resistance to shear forces). These have been extensively studied in terms of both native and modified starches, as well as their interaction with low molecular mass solutes and high molecular mass hydrocolloids. Most of all, common commercial starches such as corn, wheat, potato, cassava, or rice have been thoroughly examined [1,8,9,10]. Nevertheless, in recent years, there has been a growing interest in starches isolated from various plants, especially pulses [11], millet [12], and also less popular ones such as *Phalaris canariensis* L. or *Solanum lycocarpum* [13]. A huge effort has been made to study starch–hydrocolloids systems [14,15,16,17,18], especially xanthan [19,20,21,22], inulin [23,24], pullulan [25,26], and pectin [27,28], as well as protein [29] or amino acids [30]. Interest in mixtures of starch from various plant species is related to the expectation of possible synergistic actions of biopolymers applied as thickeners. However, the observed effects are multidirectional, and this makes it impossible not only to define specific technological recommendations but to even establish a coherent set of conclusions. The observed phenomenon depends on a lot of factors such as the botanical origin of starch, the amylose/amylopectin ratio, molecular mass distribution, the ionic/nonionic structure of the hydrocolloids, and the concentration of both polymers. Moreover, opposite effects have been observed in mixtures of the same biopolymers by different concentrations. Hence, each of the biopolymer mixtures should be analyzed and considered separately.

Studies concerning the interaction of starch with salt [31,32], acids [33], sucrose [34,35], and other sweetening agents [36,37,38] seem to be of special importance, as they are common food components. A commonly observed phenomenon is the negative effect of acids on the thickening ability of starch. Pasting and rheological parameters are significantly decreased, and the gel texture is deteriorated. These phenomena are accompanied by weakened retrogradation [33]. However, these effects can be significantly modified by the presence of salts. The salts themselves also have different effects depending on the concentration and valence of the cation. Monovalent cations are recognized as swelling inhibitors, whereas calcium chloride increases water absorption, cold paste viscosity, and the mechanical properties of starch pastes. It is related to the crosslinking effect of divalent calcium cations [32]. The effect of sucrose on the properties of starch in solutions is related to the formation of hydrogen bonds. The presence of sucrose increases gelatinization temperature and enthalpy. These phenomena are accompanied by an increase of pasting and rheological parameters, as well as the strength of gels [35]. A similar effect is also caused by the presence of polyols, i.e., maltitol, sorbitol, xylitol, and mannitol. However, it has been suggested that the molecular mass and spatial structure of the hydroxyl groups of sucrose/polyols could affect those properties by changes in the ability to form hydrogen bonds [38]. At the same time, a strong dependence of the effects on the concentration of polyols in the system is observed, which is also related to the formation of hydrogen bonds [37]. The use of a complex sweetening agent (date syrup) instead of pure chemical compounds, which are sucrose and polyols, makes the interactions more complex. It mainly regards untypical changes in pasting characteristics. A positive effect on the gelling properties is observed, which may be of industrial importance, especially in confectionery products [39].

The enormous amount of diverse information makes it difficult to come to general conclusions regarding the effect of different solutes on the rheological properties of starch. Therefore, the aim of this work is to systemize information on the rheological properties of the most popular starches in matrices of various food products, applying principal component and cluster analyses. Starches used in the study were limited to the most popular in food production, i.e., potato and corn of normal and waxy varieties. Binary mixtures containing salts (NaCl and KCl) or sweetening agents (sucrose, sorbitol, xylitol, erythritol, and glycerol) were analyzed. Moreover, the functionality of these starches in different food products, i.e., ketchup, mayonnaise, pudding, and jelly, was studied. The data presented makes it possible to compare the effect of different factors on starch rheology and may be helpful with the rational choice of thickener for food production.

## 2. Materials and Methods

### 2.1. Starch

Commercial native starches: potato (PPZ Trzemeszno, Trzemeszno, Poland), waxy potato (Avebe, Veendam, The Netherlands), corn, and waxy corn (AGRANA Starke GmbH, Aschach an der Donau, Austria) were the base working materials. All analyses were performed on a starch dry matter basis.

### 2.2. Preparation of Starch Binary Mixtures

Binary mixtures were prepared based on analytical-grade reagents and purified water (<0.1 µS/cm).

### 2.3. Preparation of Model Food Systems

Model food systems were prepared according to the following formulas:Ketchup: 60 g tomato paste (30 ± 2% dry matter), 25 g sucrose, 12 g vinegar (10%), 6.5 g starch, 3 g salt, 0.2 g finely milled black pepper, water up to 200 g.Mayonnaise (low fat): 125 g water, 120 g rapeseed oil, 17 g sucrose, 14 g starch, 10 g vinegar (10%), 10 g dried egg yolk, 6 g mustard, 3 g salt.Pudding: 20 g sucrose, 18 g starch, 250 g milk (3.8% fat).Jelly: 20 g starch, 12.5 g sucrose, 1 g citric acid, water up to 250 g.

Model food systems preparation procedures:Ketchup: All ingredients, excluding vinegar, were mixed with an overhead stirrer (R50D, Ingenieurburo CAT M. Zipperer GmbH, Germany) operating at 150 rpm. Raw ketchup samples were placed in a boiling water bath for 15 min. Vinegar and evaporated water were added at the last stage of ketchup preparation. Prepared samples were stored for 24 h at 4 °C prior to analysis [40].Mayonnaise: Prior to the preparation of the emulsion, a starch suspension was pasted along with water, sugar, and salt. The emulsion was ground in a semicircular bowl (9 cm radius) with a twin shaft mixer (400 W). Oil was added gradually (during 1 min) to avoid phase separation. Vinegar and mustard were added at the last stage of mayonnaise preparation. Ready mayonnaise was mixed for an additional 1 min. Further analyses were made on products stored for 24 h at 4 °C [41].Pudding: Starch was suspended in 50 mL of cold milk and poured into boiling milk (mixed with sucrose). Afterward, the pudding was mixed with an overhead stirrer (R50D, CAT) operating at 150 rpm and boiled for an additional 5 min. Prepared samples were allowed to set at room temperature for 1 h prior to the analysis [42].Jelly: All ingredients were mixed together with an overhead stirrer (R50D, CAT) operating at 150 rpm and placed in a boiling water bath for 15 min. Prepared samples were allowed to set at room temperature for 1 h prior to the analysis [42].

### 2.4. Determination of Pasting Characteristics

Pasting characteristics of 5% starch suspensions and binary mixtures were recorded with a Brabender Viscograh (Duisburg, Germany). Standard measurement parameters were set: measuring cartridge 0.07 Nm, a heating and cooling rate of 1.5 °C/min (within the 25–92.5–25 °C range), a holding period of 20 min at 92.5 °C. The following pasting parameters were analyzed: pasting temperature °C, final viscosity BU, breakdown BU, and setback BU.

### 2.5. Determination of Rheological Properties

Rheological properties were determined using a RheoStress1 rheometer (Haake Technik GmbH, Vreden, Germany) before the measurement samples were thermostated at 20 °C and relaxed in measuring cylinder for 5 min. Data collection and calculations were made using RheoWin 3.61 software. Measurements were performed under the following conditions: Z20 DIN Ti coaxial measurement system, 1-600-1 s^−1^ shear rate range, and time of 2 min. Obtained flow curves were described with the Ostwald de Waele equation:τ = K·γ^n^(1)
where *τ* is shear stress Pa, *K* is the consistency index Pa∙s^n^, *γ* is the shear rate s^−1^, and *n* is the flow behavior index (a dimensionless number that indicates the closeness to Newtonian flow).

### 2.6. Determination of Universal Texture Profile

Universal texture profiles (Texture Profile Analysis) were determined with a TA-XT2 texturometer (Stable Micro Systems, Godalming, UK). A standard “double bite test” was performed with an aluminum cylindrical head (35 mm diameter) in a 68-mm diameter vessel on a depth of 20 mm with a speed of 0.5 mm/s. Hardness (N), adhesiveness (N∙s), cohesiveness, springiness, and gumminess (N) were determined.

### 2.7. Statistical Analysis

All analyses were performed in triplicate, and the results are presented as mean ± standard deviation. Experimental data was studied using a one-way analysis of variance and Tukey’s post hoc test. Principal component analysis (PCA) was performed based on the correlation matrix. Cluster analysis (CA) was conducted based on Ward’s method, and the Euclidean distance was used as a measure of similarity. The statistical analyses were performed using Statistica 13.3 software (Dell Software Inc., Round Rock, TX, USA).

## 3. Results and Discussion

### 3.1. Starch Pastes

The characterization of pasting profiles is one of the most useful methods for the evaluation of the functionality of starch preparations in food technology. Simple viscographic analysis can provide basic information regarding gelatinization temperature, viscosity, as well as rheological stability [1,8,43]. The pasting parameters of investigated starch suspensions are presented in Table 1. Native tuber starches and waxy varieties, regardless of botanical origin, are believed to reveal a high type of swelling characteristics. The case of the investigated preparations is mainly reflected in the high values of the breakdown parameter, especially when compared to final viscosity. Corn starch was characterized by a medium type of swelling characteristics, indicated by a lack of breakdown and a moderate value of setback (compared to final viscosity), which is typical for this variety. In food technology, the most important parameter related to viscographic analysis is the paste’s final viscosity, which reflects the ability to thicken food products. The highest values of these parameters were recorded, respectively, for normal potato, waxy potato, waxy corn, and, lastly, for normal corn starch. These results are consistent with the previous results reported by various authors [1,44,45,46]. However, more accurate data regarding the rheological properties of starch pastes can be derived from the Ostwald de Waele equation parameters, i.e., the consistency index and the flow behavior index [47,48,49] that can be found in Table 1. The employed equation was very well fitted to the experimental data as values of the coefficient of determination (R^2^) exceeded 0.98. Both normal starch varieties were characterized by higher values of the consistency index compared to their waxy counterparts. This phenomenon could be associated with the lack of gelling properties of waxy starches, which is related to the lack of amylose, thus resulting in relatively low viscosity of these preparations, especially at low shear rates. The convergence to Newtonian flow indicated by the flow behavior index was similar for all samples characterized by a high type of swelling characteristics. The significantly lower value of this parameter recorded for normal corn starch is the result of low resistance to higher shear forces, which can be related to the strong gelling properties of this preparation. Similar observations regarding the differences between normal and waxy varieties of the same botanical origin can be drawn from the analysis of universal texture profiles (Table 1). Normal starch varieties, which tend to form gel structures, were characterized by higher absolute values of hardness, adhesiveness, and gumminess, indicating a more solid-like structure of these pastes. In contrast, waxy varieties were behaving more fluid-like, which was reflected in the lower values of the texture profile parameters, as well as higher values of the flow behavior index.

The differences and similarities between the rheological and texture properties of investigated pastes were also confirmed by PCA analysis (Figure 1). Two first principal components explained over 99.3% of total variance for the data presented in Table 1. The PCA score plot (Figure 1A) highlights the similar properties of both waxy starch varieties, which were mostly influenced by comparable and relatively high values of the flow index and adhesiveness. In contrast, normal varieties were proven to have more unique characteristics that were mostly influenced by rheological and texture parameters, which tend to increase as a result of sol–gel transition, i.e., consistency index, final viscosity, or hardness. PCA loadings (Figure 1B) indicate several correlations between the different texture parameters of the analyzed pastes. Firstly, a positive correlation between adhesiveness and springiness could be observed. Secondly, a positive correlation for hardness and gumminess (the product of hardness and cohesiveness) should be noted. Lastly, both of these two groups were negatively correlated. Interestingly, a weak correlation (calculated Spearman rank coefficient 0.400; *p* < 0.05) was observed between the consistency index and final viscosity, which should be attributed to the difference in shear forces applied in those studies.

Hierarchical cluster analysis (Figure 2) strengthened the thesis regarding the similar properties of waxy potato and waxy corn starch varieties and the more unique properties of the normal ones. However, normal corn starch stands out the most for several reasons, including pasting and rheological and texture properties. Concerning food technology, normal corn starch is characterized by a significantly higher gelatinization temperature and low final viscosity, the most non-Newtonian flow, and higher values of texture parameters compared to the other three preparations. In contrast, normal potato starch differentiates only by the high values of the parameters relating to viscosity, i.e., final viscosity, consistency index, and hardness.

### 3.2. Binary Mixtures

#### 3.2.1. Salts

Sodium chloride is used extensively in food technology as an ingredient that improves the sensory perception of a vast majority of food products. As a result, many populations consume significantly more salt than the World Health Organisation recommendation. Unfortunately, high sodium diets are associated with the risk of cardiovascular diseases, especially related to high blood pressure [50]. Due to the ability of potassium chloride to regulate blood pressure, it is considered as a replacement for products designed for a low sodium diet. Both of these electrolytes can fundamentally influence the physical transformation of starch during food processing (heat treatment), regardless of the significant influence of both of these compounds on pasting and rheological and texture properties of the investigated starch pastes (Appendix A). The increase of pasting temperature and decrease of consistency index with the accompanying increase in convergence with the Newtonian flow of pastes should be primarily mentioned among the observed effects. The observed phenomena were similar regardless of the type of salt used. Similar observations regarding both the influence on rheological behavior and differences between the type of salt used were reported for lotus root starch–konjac glucomannan ternary mixtures with sodium, potassium, as well as calcium chloride [33].

Principal component analysis performed for starch pastes containing salts (Figure 3) shared many similarities with the analysis performed for pure pastes (Figure 1). The first two components explained 78.6% of the total variance between the samples containing salts. Despite the narrower data representation, the correlations between the rheological parameters indicated by the loadings plot remained almost unchanged. The only differences that were visible were for breakdown and gumminess, which were negatively correlated with each other. The observed change in breakdown may be the result of the flattening of the gelatinization curve caused by the presence of salts. In contrast, changes in gumminess could be related to the deterioration of texture. As this parameter is a product of hardness and cohesiveness, any changes in the overall texture profile are especially reflected in it.

The PCA score plot (Figure 3A) shows that normal starch varieties were the most influenced by the presence of salting agents. Moreover, only minor differences were observed between pastes with different cation types (within the same preparation). Finally, waxy starch varieties could be undoubtedly assigned to one set (green ellipse).

The cluster analysis dendrogram for starch–salt binary mixtures (Figure 4) showed many similarities to those prepared for pure starch pastes (Figure 2). Firstly, the addition of salt did not affect the initial clustering of the samples. One could clearly differentiate the three main clusters containing corn starch, waxy starches, and potato starch pastes. This indicates that although both sodium and potassium chloride influence the rheological properties of starch pastes, their effect is not as relevant as the botanical origin or the variety of starch preparation. These observations correlate to the conclusions resulting from the PCA score plot (Figure 3A). The only difference between those two studies (PCA and CA) is the relationship between distances of different waxy starch varieties containing potassium chloride. The Euclidean distance presented in the PCA score plot between these samples is closer to other pastes of the same botanical origin and variety instead of each other, as in the cluster analysis.

Interestingly, the effect of the cation type was dependent on the presence of amylose. In the case of potato and corn starches, the influence of the presence of salt on clustering was obvious—binary mixtures were more closely linked compared to pure starch paste. However, in the case of waxy starch varieties, binary mixtures containing potassium chloride were more closely linked than any other sample. This phenomenon can be partially related to differences in the influence of potassium and sodium ions on starch pastes. However, this was predominantly a result of a much lower impact of salts on the rheological properties of amylopectin-only starches compared to normal ones.

#### 3.2.2. Sucrose and Polyols

Sucrose has been, traditionally, the most popular sweetener. However, the negative effects of excess sucrose consumption, such as type 2 diabetes, obesity, and poor oral health, has led to the replacement of sugar by sugar alcohols [51]. Sugar and sugar alcohols are believed to be kosmotropic substances that stabilize the native structure of biopolymers, including starch [52]. The most common views on the role of these substances in starch–polyol systems, as a reason for the changes in the viscosity of these systems, indicate some kind of “competition” in access to water that is necessary for the process of starch gelatinization and conformational changes affecting the degree of intermolecular starch association [53]. Moreover, changes in the gelatinization process are related to a much weaker plasticizing effect of sweeteners, as compared to water. This results in an increase of the gelatinization temperature of the starch–polyol system. It should be emphasized that the complexity of the chemical structure of sugar and sugar alcohols plays a key role in the observed phenomena [54,55]. In the case of studied sweetening substances, the effects on starch properties were not so unequivocal as those of salts. Both the increase and decrease of gelatinization temperature was observed due to the addition of various sweeteners. Moreover, both an increase and a decrease in the consistency coefficient occurred as a result of the addition of sugar and sugar alcohols. Similar variability was also observed for other rheological parameters (Appendix A).

PCA plots for starch pastes containing sugar and polyols are presented in Figure 5. They share many similarities with both those presented for plain pastes (Figure 1) and pastes with the addition of salts (Figure 3). The first two plotted components explain over 85.8% of the total variance. The loadings plot remains almost unchanged compared to that of pure pastes. The main reason behind that phenomenon is the fact that the addition of all the investigated sweetening substances did not substantially change the course of the gelatinization curve. Moreover, changes in TPA and the rheological properties related to the formation of stronger texture were similar for all types of starch preparations. Waxy starch varieties formed, on a score plot, two closely positioned data sets based on their botanical origin. In contrast, the data points of normal starch varieties were more scattered, especially when considering samples containing sucrose. The distancing of those binary mixtures in both studies should be attributed to the more significant influence of sucrose on the formation of “strong” paste texture.

The influence of the investigated polyols on clustering (Figure 6) shared many similarities with those of salts (Figure 4). Firstly, once again, a clear separation of groups containing the same starch preparation was observed. Secondly, waxy starch varieties were relatively closely linked to each other. Finally, no clear relationship between the type of polyol used and clustering was present. However, several patterns should be noted, especially in the case of normal starch varieties. Quite obviously, binary mixtures of those starches containing sugar alcohols were the most linked with each other. Surprisingly, the next closest linkage was to pure starch paste and then, subsequently, to binary mixtures with sucrose. This indicates that although the strength of the impact of sucrose and sugar alcohols on starch pastes is quite similar, the course of the changes is related to the type of cosolute. The aforementioned phenomenon was not observed for waxy starch varieties or dependent on the molecular weight of the cosolute used.

### 3.3. Food Products

Food products are a complex matrix of a vast amount of substances that may significantly influence starch-thickening properties (Appendix A). Therefore, it should not come as a surprise that the clustering of ketchup, mayonnaise, pudding, and jelly (Figure 7) did not resemble any of the dendrograms presented for starch pastes or their binary mixtures.

The most coherent group regarding different food products was observed for ketchup (brick-red color), although mayonnaise thickened with waxy corn starch was also included in it. The most similar products in this group were, predictably, ketchup thickened with waxy starches. The similarity of mayonnaise with waxy corn starch to ketchup in this study should not be a surprise, as commercially prepared low-fat mayonnaises are usually thickened with a chemically modified variant of this starch. Despite the fact that mayonnaise is an emulsion and ketchup is a suspension type of sauce, their sensory mechanical properties are quite similar. In turn, the larger distance from this cluster, towards other mayonnaises, may have resulted from undesirable changes in texture that are caused by other starch variants, which are not recommended for this type of product as they may develop a gelatinous texture [41].

Jelly-type deserts were divided into two clusters. One small cluster contained only jellies prepared with waxy starches (dark green). A second larger cluster, apart from jellies prepared with normal starch varieties, included puddings thickened witch waxy starches (orange). The observed clustering is the effect of two factors: gelation of jellies containing starches with amylose and the formation of pudding curd. Eventually, the differences with the last cluster containing pudding prepared with normal starches (yellow) may be attributed to the synergism of amylose and milk proteins, resulting in the formation of strong pudding curd.

The food product clustering was consistent with the data presented for the PCA score plot (Figure 8A). However, differences between the designated sets were less evident. Similar to CA, the biggest set included all the investigated ketchups and mayonnaises thickened with both variants of corn starch. The other indicated sets also coincided with CA but were small—containing two to three products. Lastly, the most diverse products were also the same, i.e., puddings thickened with normal and waxy starch varieties and jelly thickened with normal corn starch.

Contrary to the score plot, the presented loadings (Figure 8B) provide new and significant data, especially when considering the PCA performed for binary mixtures. Firstly, no correlation between universal texture profiles and flow parameters was observed. This indicates that both TPA and the rheological studies provide different information regarding food product samples and cannot be used interchangeably. Secondly, clear but interesting correlations between all rheological parameters, as well as all texture parameters, should be noted. Regarding rheological properties, the increase of viscosity (indicated by consistency index) results in the formation of stronger thixotropic structures and more non-Newtonian flow behavior. In turn, the formation of stronger (harder) texture results in higher gumminess and lower adhesiveness, cohesiveness, and springiness.

The aforementioned observations regarding the different types of food products demonstrate the complicated task of technologists in choosing proper thickener. The complexity of food colloidal systems and the vast amount of interactions hinder the selection of the right solution, which is perfectly shown by the dendrogram presented in Figure 7, as well as the PCA score plot in Figure 8A. Moreover, this proves that the proper selection of starch as a thickener for food products based on data regarding model systems is almost impossible.

## 4. Conclusions

The functionality of native starches in food systems is a result of numerous factors that influence the efficiency of the preparation and quality of the final product. Native potato and corn starches are significantly different in their physicochemical properties, resulting in different areas of application for those preparations. This conclusion is only true for the normal varieties, whereas waxy starch varieties are characterized by similar pasting and rheological and texture properties. Moreover, waxy starch pastes are less altered by the presence of cosolutes, i.e., salts and sugar alcohols. This may potentially result in a wider area of application.

A comparison of the studies performed on food product matrices and model binary systems showed the low usefulness of the latter in the quality design of food products. Nevertheless, starch binary systems were proven to be useful material for industrial screening tests.

## Figures and Tables

**Figure 1 foods-09-01073-f001:**
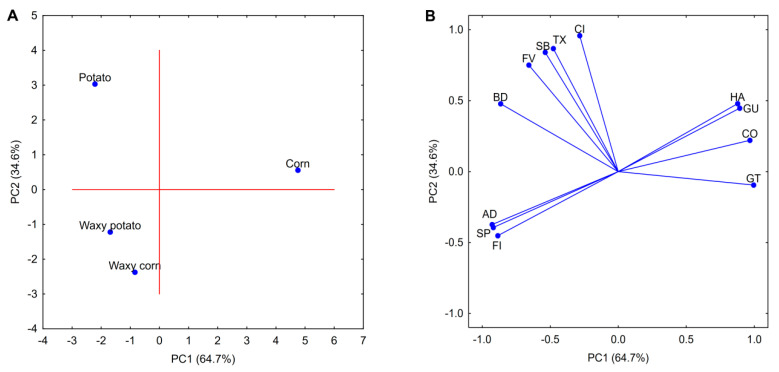
Principal component analysis (PCA) score (**A**); loadings plot (**B**); rheological and texture properties of 5% native starch pastes. Explanatory notes: GT—gelatinization temperature; FV—final viscosity; BD—breakdown; SB—setback; CI—consistency index; FI—flow behaviour index; TX—thixotropy; HA—hardness; AD—adhesiveness; CO—cohesiveness; SP—springiness; GU—gumminess.

**Figure 2 foods-09-01073-f002:**
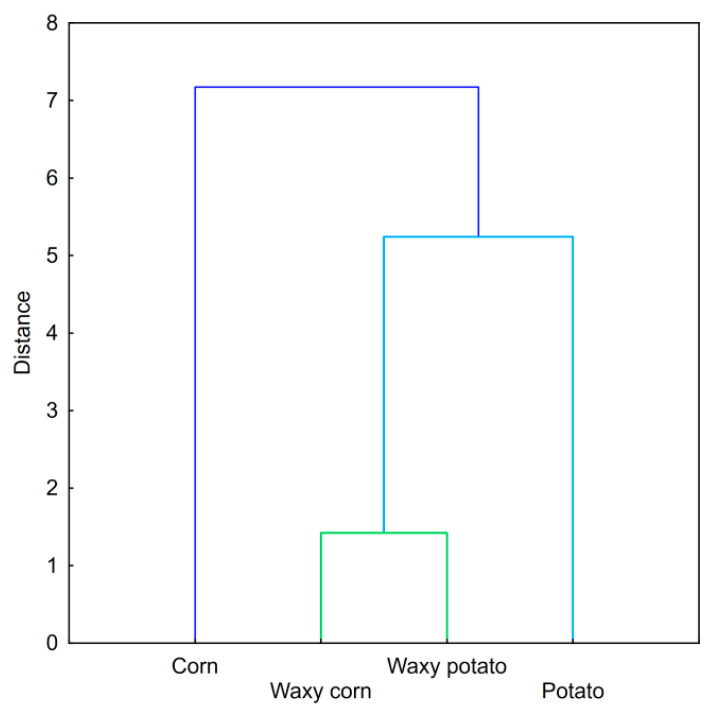
Dendrogram; hierarchical cluster analysis of 5% native starch pastes.

**Figure 3 foods-09-01073-f003:**
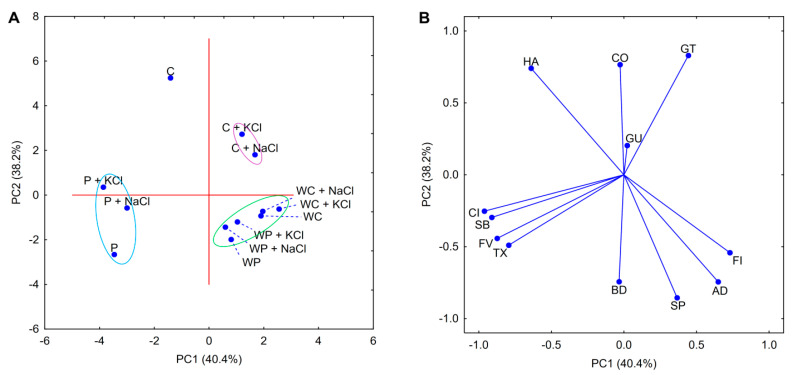
PCA score (**A**); loadings plot (**B**); rheological and texture properties of binary mixtures of 5% native starch pastes with 3% salt. Explanatory notes: P—potato starch; WP—waxy potato starch; C—corn starch; WC—waxy corn starch; NaCl—sodium chloride; KCl—potassium chloride; GT—gelatinization temperature; FV—final viscosity; BD—breakdown; SB—setback; CI—consistency index; FI—flow behavior index; TX—thixotropy; HA—hardness; AD—adhesiveness; CO—cohesiveness; SP—springiness; GU—gumminess.

**Figure 4 foods-09-01073-f004:**
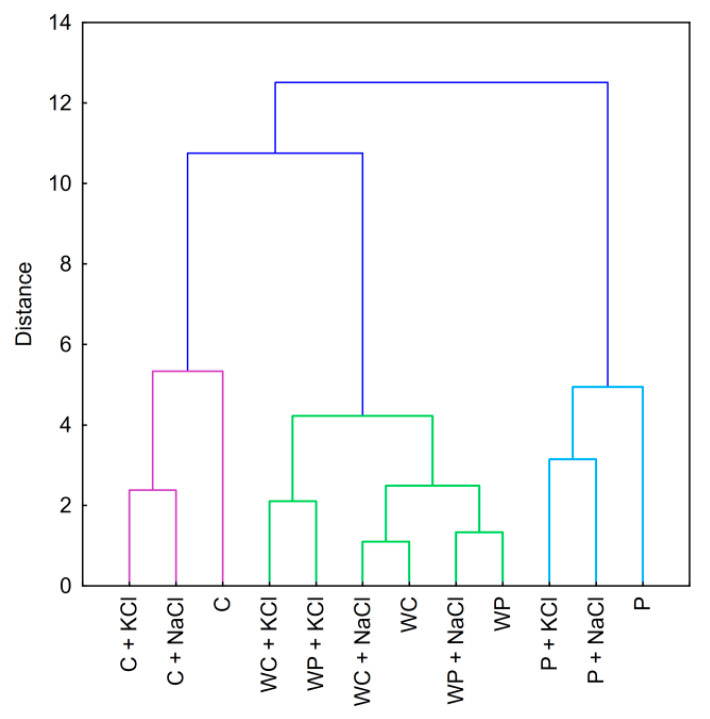
Dendrogram of hierarchical cluster analysis of binary mixtures of 5% native starch pastes with 3% salt. Explanatory notes: P—potato starch; WP—waxy potato starch; C—corn starch; WC—waxy corn starch; NaCl—sodium chloride; KCl—potassium chloride.

**Figure 5 foods-09-01073-f005:**
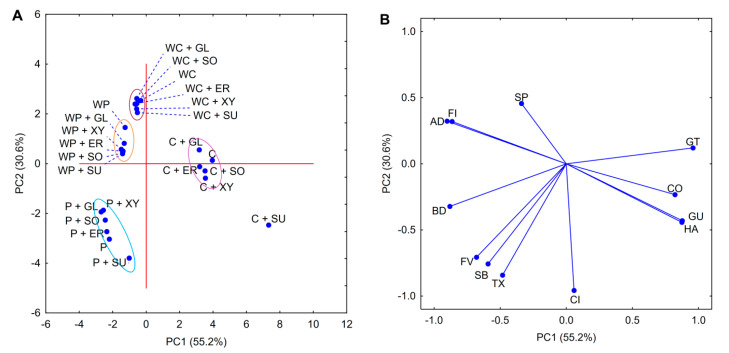
PCA score (**A**); loadings plot (**B**); rheological and texture properties of binary mixtures of 5% native starch pastes with 10% sweetening agents. Explanatory notes: P—potato starch; WP—waxy potato starch; C—corn starch; WC—waxy corn starch; SU—sucrose; SO—sorbitol; XY—xylitol; ER—erythritol; GL—glycerol; GT—gelatinization temperature; FV—final viscosity; BD—breakdown; SB—setback; CI—consistency index; FI—flow behavior index; TX—thixotropy; HA—hardness; AD—adhesiveness; CO—cohesiveness; SP—springiness; GU—gumminess.

**Figure 6 foods-09-01073-f006:**
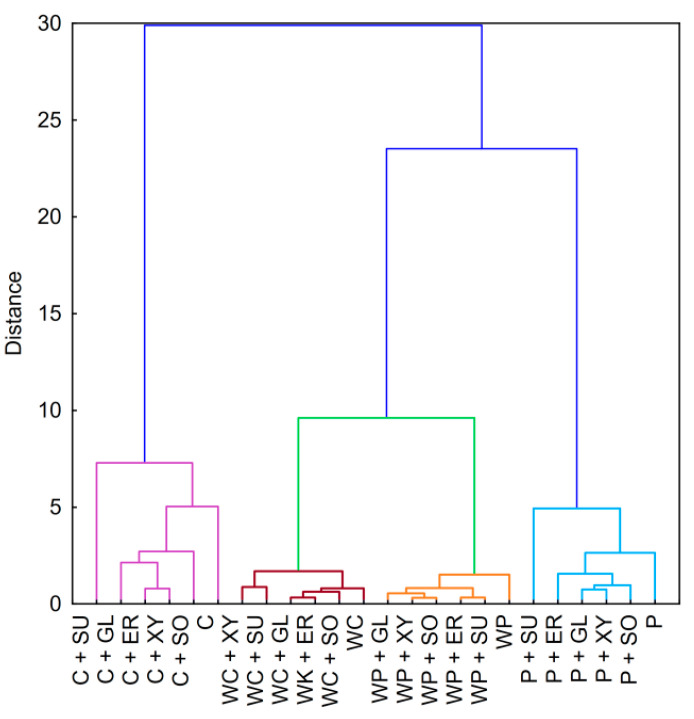
Dendrogram of hierarchical cluster analysis of binary mixtures of 5% native starch pastes with 10% sweetening agents. Explanatory notes: P—potato starch; WP—waxy potato starch; C—corn starch; WC—waxy corn starch; SU—sucrose; SO—sorbitol; XY—xylitol; ER—erythritol; GL—glycerol.

**Figure 7 foods-09-01073-f007:**
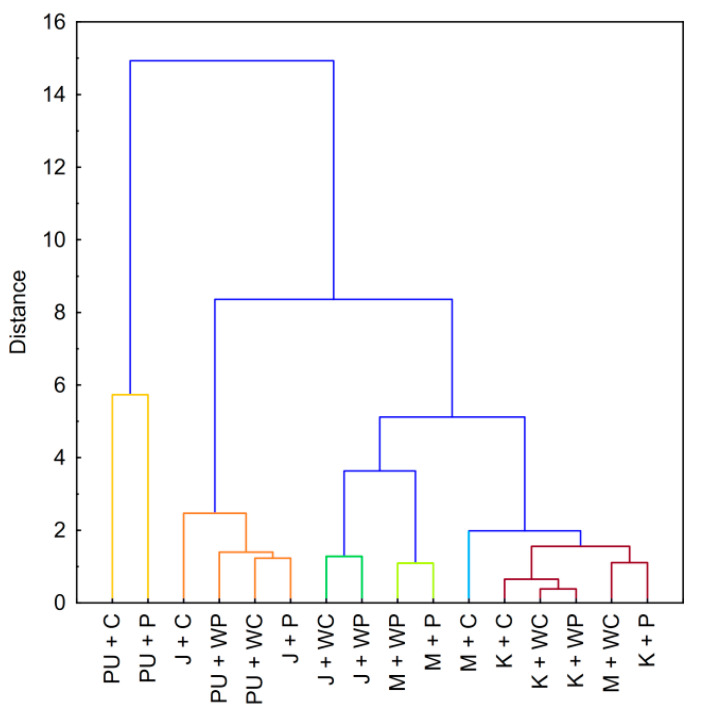
Dendrogram of hierarchical cluster analysis of food products thickened with native starch pastes. Explanatory notes: P—potato starch; WP—waxy potato starch; C—corn starch; WC—waxy corn starch; K—ketchup; M—mayonnaise; PU—pudding; J—jelly.

**Figure 8 foods-09-01073-f008:**
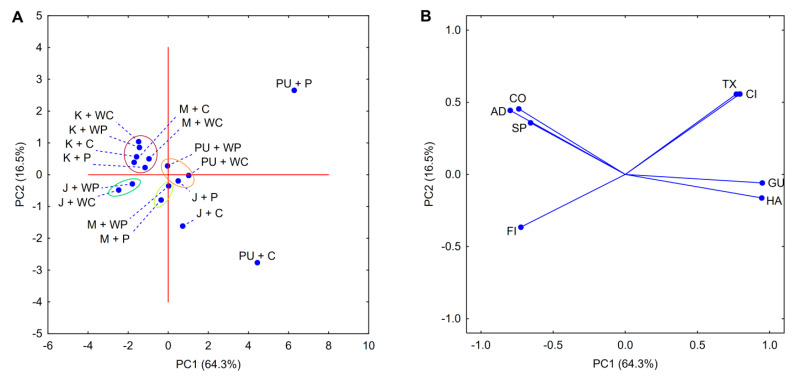
PCA score (**A**); loadings plot (**B**); rheological and texture properties of food products thickened with native starch pastes. Explanatory notes: P—potato starch; WP—waxy potato starch; C—corn starch; WC—waxy corn starch; K—ketchup; M—mayonnaise; PU—pudding; J—jelly; CI—consistency index; FI—flow behavior index; TX—thixotropy; HA—hardness; AD—adhesiveness; CO—cohesiveness; SP—springiness; GU—gumminess.

**Table 1 foods-09-01073-t001:** Rheological properties and texture parameters of 5% native starch pastes.

Starch	Potato	Waxy Potato	Corn	Waxy Corn
Gelatinization temperature (°C)	62.0 ± 0.1	64.8 ± 0.1	84.5 ± 0.1	68.7 ± 0.1
Final viscosity (BU)	1133 ± 24	612 ± 15	296 ± 10	348 ± 9
Breakdown (BU)	1319 ± 30	888 ± 8	6 ± 1	422 ± 7
Setback (BU)	621 ± 20	233 ± 10	149 ± 7 ^a^	140 ± 10 ^a^
Consistency index (Pa∙s^n^)	31.86 ± 0.98	10.93 ± 0.60 ^ab^	16.71 ± 5.09 ^b^	5.47 ± 0.36 ^a^
Flow behavior index (-)	0.468 ± 0.001 ^a^	0.495 ± 0.006 ^a^	0.356 ± 0.044	0.508 ± 0.006 ^a^
Thixotropy (Pa∙s^−1^)	49,535 ± 2496	10054 ± 547 ^a^	7208 ± 3298 ^a^	5503 ± 839 ^a^
Hardness (N)	0.42 ± 0.03 ^a^	0.35 ± 0.00 ^a^	0.57 ± 0.07	0.34 ± 0.00 ^a^
Adhesiveness (N∙s)	−0.21 ± 0.06	0.00 ± 0.01 ^a^	−0.92 ± 0.13	0.00 ± 0.00 ^a^
Cohesiveness (-)	0.74 ± 0.01 ^a^	0.73 ± 0.00 ^a^	0.78 ± 0.04 ^a^	0.74 ± 0.00 ^a^
Springiness (-)	0.99 ± 0.01 ^a^	1.00 ± 0.00 ^a^	0.96 ± 0.04 ^a^	1.00 ± 0.00 ^a^
Gumminess (N)	0.31 ± 0.02 ^a^	0.26 ± 0.00 ^a^	0.44 ± 0.07	0.25 ± 0.00 ^a^

Explanatory notes: data are expressed as mean value ± standard deviation. Values denoted with the same letter within a row (^a^, ^b^) do not differ significantly (*p* > 0.05).

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
