# Peer review of "Functionality of Native Starches in Food Systems: Cluster Analysis Grouping of Rheological Properties in Different Product Matrices"

_foods, 2020, doi:10.3390/foods9081073_

Round 1
Reviewer 1 Report
The manuscript reports a study on the investigation of rheological and texture properties of waxy corn and potatoes and binary mixtures containing salts or sweetening agents. Furthermore, the relationship between starch and physical properties was investigated using PCA. The results of this study will be important data for the field related to starch rheology. The manuscript is well organized and written.
Lines 11-24 Need to complete the abstract paragraph. Reduce the introduction section and write down the important results data from this study and write your final conclusion.
Lines 98-100. Please write the moisture content of the starch.
Lines 141-144. Please write the size or volume of the measured sample.
Lines 153-179. Many researchers report a lot of results on pasting properties. Please compare the results of the experiment in Table 1. with the results of the previous paper.
Lines 213-223. Only general salt is described. There is insufficient discussion about the research results. Please compare the results with other researchers' results.
Author Response
Thank you for your your valuable comments and remarks. We have revised the manuscript according to your and other reviewers suggestions. All changes were made using track changes mode.
Review: The manuscript reports a study on the investigation of rheological and texture properties of waxy corn and potatoes and binary mixtures containing salts or sweetening agents. Furthermore, the relationship between starch and physical properties was investigated using PCA. The results of this study will be important data for the field related to starch rheology. The manuscript is well organized and written.
Response: Thank you.
Review: Lines 11-24 Need to complete the abstract paragraph. Reduce the introduction section and write down the important results data from this study and write your final conclusion.
Response: Abstract was extended by final conclusion (Line 25-26), style was improved in various places. The introduction section of abstract remained the same to maintain logical narration.
Review: Lines 98-100. Please write the moisture content of the starch.
Response: Information regarding dry matter of starch was placed in materials and methods section.
Review: Lines 141-144. Please write the size or volume of the measured sample.
Response: Diameter of measuring vessel was provided (Line 155), instead of sample size due to the fact that always excess volume must be provided.
Review: Lines 153-179. Many researchers report a lot of results on pasting properties. Please compare the results of the experiment in Table 1. with the results of the previous paper.
Response: 4 citations consistent with presented data were provided (Line 179)
Review: Lines 213-223. Only general salt is described. There is insufficient discussion about the research results. Please compare the results with other researchers' results.
Response: Clarification regarding different salts was placed. Citation with comparison to literature data was provided (Line 240-243).
Reviewer 2 Report
The authors investigated the rheological properties of starches in matrices of various food products using principal component and cluster analysis. They found that waxy starches showed similar rheological properties in all food models compared to normal varieties. One waxy starch variety could be substituted by others, without any significant impact on the rheological properties and texture of the food product. Moreover, waxy starch preparations were less altered by the presence of cosolutes. Overall, this is a well-organized research article with some interesting findings.
Materials and Methods should be divided into several subsections for an easy reading.
The language should be substantially improved by correcting all grammatical errors and typos.
Author Response
Thank you for your your valuable comments and remarks. We have revised the manuscript according to your and other reviewers suggestions. All changes were made using track changes mode.
Review: The authors investigated the rheological properties of starches in matrices of various food products using principal component and cluster analysis. They found that waxy starches showed similar rheological properties in all food models compared to normal varieties. One waxy starch variety could be substituted by others, without any significant impact on the rheological properties and texture of the food product. Moreover, waxy starch preparations were less altered by the presence of cosolutes. Overall, this is a well-organized research article with some interesting findings.
Response: Thank you.
Review: Materials and Methods should be divided into several subsections for an easy reading.
Response: Materials and methods were divided into 7 subsections.
Review: The language should be substantially improved by correcting all grammatical errors and typos.
Response: Extensive improvement of introduction was performed along with minor corrections in whole manuscript.
Reviewer 3 Report
In my opinion, the manuscript entitled "Functionality of native starches in food systems. Cluster analysis grouping of rheological properties in different product matrices " is very well written .
- Please replace "hrs" by "h"
- Please replace "s-1" by "s-1"
- Did you carry out oscillating tests, such as stress sweeps or frequency sweeps?.
- Concerning the use of Oswald de Waele model: You can compare between samples if you modify the model, becayse consistency index units depend on flow index. For example, see references:
- Trujillo-Cayado, L. A., Santos, J., Alfaro, M. C., Calero, N., & Muñoz, J. (2016). A further step in the development of oil-in-water emulsions formulated with a mixture of green solvents. Industrial & Engineering Chemistry Research, 55(27), 7259-7266
- Table 1: I think that some Tukey´s analysis letters are missing.
- Maybe the inclusion of at least one comparison of some flow curves could help the reader to understand the rheological behaviour of these samples.
Response: In cream viscosity curves, major differences were observed at higher shear rate values. For that reason, we chose to maintain the logarithmic-linear scale representation. However, this information was added in the manuscript. Please consult lines 226 and 227.
Figure 2 in log-lin scale representation:
Figure 2 in log-log scale representation:
- Concerning this sentence: "The K gives an indication about sample viscosity. However, to be properly compared, the materials should present similar n [5,6,54-58]." You can compare between samples if you modify the model. For example, see references:
- Trujillo-Cayado, L. A., Santos, J., Alfaro, M. C., Calero, N., & Muñoz, J. (2016). A further step in the development of oil-in-water emulsions formulated with a mixture of green solvents. Industrial & Engineering Chemistry Research, 55(27), 7259-7266
ç
Line 114 (and all manuscript)
Author Response
Thank you for your your valuable comments and remarks. We have revised the manuscript according to your and other reviewers suggestions. All changes were made using track changes mode.
Review: In my opinion, the manuscript entitled "Functionality of native starches in food systems. Cluster analysis grouping of rheological properties in different product matrices " is very well written .
Response: Thank you.
Review:
- Please replace "hrs" by "h"
- Please replace "s-1" by "s-1"
- Did you carry out oscillating tests, such as stress sweeps or frequency sweeps?.
- Concerning the use of Oswald de Waele model: You can compare between samples if you modify the model, becayse consistency index units depend on flow index. For example, see references:Trujillo-Cayado, L. A., Santos, J., Alfaro, M. C., Calero, N., & Muñoz, J. (2016). A further step in the development of oil-in-water emulsions formulated with a mixture of green solvents. Industrial & Engineering Chemistry Research, 55(27), 7259-7266
- Table 1: I think that some Tukey´s analysis letters are missing.
- Maybe the inclusion of at least one comparison of some flow curves could help the reader to understand the rheological behaviour of these samples.
- Concerning this sentence: "The K gives an indication about sample viscosity. However, to be properly compared, the materials should present similar n [5,6,54-58]." You can compare between samples if you modify the model. For example, see references: Trujillo-Cayado, L. A., Santos, J., Alfaro, M. C., Calero, N., & Muñoz, J. (2016). A further step in the development of oil-in-water emulsions formulated with a mixture of green solvents. Industrial & Engineering Chemistry Research, 55(27), 7259-7266
- Line 114 (and all manuscript)
Response:
- Changed hrs to h in whole manuscript.
- Changed formatting where applicable.
- We did perform frequency sweep tests for pure pastes only. Obtained results did not contribute substantially to add value to presented data, therefore this experiment was suspended.
- Ostwald de Waele model was used as it is the most commonly used one for starch pastes in the literature. The main reasons behind usage of that model was simplicity and clear information for the reader. Several modifications of power law equation are possible and used in literature when desired/necessary. However in case of submitted manuscript our mail goal was to differentiate starch using PCA and CA based on most commonly used data. Therefore mentioning of K as viscosity was only referenced to low shear rates, to which it is applicable. Employment of modified model, would not substantially influence clustering of the samples.
- All superscripted letters were checked and are correct, only samples do not differing significantly were indicated.
- We did not decide to place flow curves to maintain visual clarity of the manuscript. The main focus of the manuscript was PCA and CA. Basic rheological behavior of investigated starches is well known, often and widely published in literature. To facilitate potential reader we placed information about values of coefficient of determination (Line182-183) which prove good adjustment of the model to the experimental data.
- Response consistent with answer in point 4.
Changed minutes to min in whole manuscript.